# Acute Effects of High Doses of Caffeine on Bar Velocity during the Bench Press Throw in Athletes Habituated to Caffeine: A Randomized, Double-Blind and Crossover Study

**DOI:** 10.3390/jcm10194380

**Published:** 2021-09-25

**Authors:** Aleksandra Filip-Stachnik, Michal Krzysztofik, Juan Del Coso, Michal Wilk

**Affiliations:** 1Institute of Sport Sciences, Jerzy Kukuczka Academy of Physical Education, 40-065 Katowice, Poland; m.krzysztofik@awf.katowice.pl (M.K.); m.wilk@awf.katowice.pl (M.W.); 2Exercise Physiology Laboratory, Universidad Rey Juan Carlos, 28942 Fuenlabrada, Spain; juan.delcoso@urjc.es

**Keywords:** resistance exercise, upper limbs, adverse effects, ergogenic substances, sports performance

## Abstract

Chronic intake of caffeine may produce a reduction in the potential performance benefits obtained with the acute intake of this substance. For this reason, athletes habituated to caffeine often use high doses of caffeine (≥9 mg/kg) to overcome tolerance to caffeine ergogenicity due to chronic intake. The main objective of the current investigation was to evaluate the effects of high caffeine doses on bar velocity during an explosive bench press throw in athletes habituated to caffeine. Twelve resistance-trained athletes, with a moderate-to-high chronic intake of caffeine (~5.3 mg/kg/day) participated in a randomized double-blind and randomized experimental design. Each participant performed three identical experimental sessions 60 min after the intake of a placebo (PLAC) or after the intake of 9 (CAF-9) or 12 mg/kg (CAF-12) of caffeine. In each experimental session, the athletes performed five sets of two repetitions of the bench press throw exercise with a load equivalent to 30% of their one-repetition maximum. In comparison to PLAC, the intake of caffeine increased peak and mean velocity (*p* < 0.01) during the five sets of the bench press throw exercise. There were no significant differences in peak and mean bar velocity between the two doses of caffeine (CAF-9 vs. CAF-12; *p* = 0.91, = 0.96, respectively). The ingestion of high doses of caffeine was effective in producing an increase in mean and peak bar velocity during the bench press throw in a group of habitual caffeine users. However, using CAF-12 did not offer additional benefits for performance with respect to CAF-9.

## 1. Introduction

Caffeine is the world’s most consumed psychoactive drug and is widely used as a ergogenic aid in competitive sports [1]. Recently, ergogenicity of caffeine for several forms of resistance exercise (i.e., mainly muscle strength and strength-endurance) has been confirmed and summarized in several systematic reviews and meta-analyses [2,3,4]. There is consensus that the main mechanism explaining the ergogenic effect of caffeine for resistance exercise is the action of caffeine as an antagonist of adenosine receptors, promoting an elevated release of neurotransmitters [5]. There is also evidence based on investigations testing the effect of caffeine on isolated muscles under in vitro conditions to suggest that, under physiological concentrations of caffeine, this substance can potentiate skeletal muscle force and muscle power, which can contribute to the overall ergogenic effect of caffeine on resistance exercise [6]. Collectively, it seems that caffeine’s ergogenicity is obtained through a number of mechanisms that work synergistically to promote the performance-enhancing effect obtained with the acute consumption of this substance. Nevertheless, the caffeine dose required to obtain a direct effect on the skeletal muscle is greater than that needed to block adenosine receptors [7].

Doses of caffeine in the range of 3 to 6 mg/kg of body mass are the most commonly used to induce acute ergogenic effect of caffeine on resistance exercise because they are effective in producing benefits in several exercise and muscle performance variables [1]. However, higher doses of caffeine are often used by some athletes with the aim of augmenting the benefits obtained from normal doses of caffeine [8,9]. From a physiological point of view, the use of high doses of caffeine is not supported by evidence, as increased doses of caffeine beyond the “normal” 3–6 mg/kg-dose fail to elicit further positive responses on performance [8,9]. However, chronic caffeine intake may produce tolerance to the ergogenic effect of caffeine [10,11] due to the new creation of binding sites for adenosine [12]. Hence, individuals with habituation to caffeine may need doses up to their habitual caffeine intake to obtain benefits from acute caffeine intake [13]. In several previous studies, high doses of caffeine (from 9 to 13 mg/kg) have shown a positive effect of acute caffeine intake on several forms of physical performance [14,15,16,17,18,19,20]. However, only three of these studies analyzed the impact of high doses of caffeine (i.e., 9 and 11 mg/kg) on performance during resistance exercise [14,15,16] and showed conflicting results. In the study by Pallarés et al. [16], 9 mg/kg of caffeine improved bar velocity and power output during bench press and full squat exercises with increasing loads from 25% to 90% of the one-repetition maximum (1RM). Wilk et al. [15] also found a positive effect of 9 and 11 mg/kg of caffeine to increase 1RM in the bench press exercise, but both doses failed to enhance power output and bar velocity during a strength endurance test. It should be noted that such high doses of caffeine in these three studies [14,15,16] led to a high frequency of side effects typically associated with caffeine intake, which showed a dose-dependent prevalence.

With this background, the use of high doses of caffeine may be an effective supplementation protocol for some athletes with high habituation to this substance through chronic intake. There is evidence showing that some individuals can consume high doses of caffeine (up to 600–800 mg/day) without experiencing such effects as tachycardia, headache or anxiety [21]. This is because the intensity and type of physiological responses to caffeine are known to vary among individuals due to differences in pharmacokinetics, pharmacodynamics, and possible tolerance [21,22]. While such doses of caffeine seem extreme and unnecessary, in some research analyzing the impact of caffeine for habitual users, the mean daily level of caffeine consumption reached these levels [23,24,25]. Furthermore, urine caffeine concentration higher than 12 μg/mL (which is obtained after intake of ~10 mg/kg of caffeine) were found in several samples of athletes tested for doping control [8,9,26]. These data suggest that the use of high doses of caffeine exists among athletes, while the most probable cause to explain this practice is the need to use high doses of caffeine to overcome the tolerance induced by chronic intake [10,11,23]. Therefore, the aim of this study was to evaluate the acute effect of 9 and 12 mg/kg of caffeine on bar velocity changes in resistance-trained athletes habituated to caffeine during a bench press throw (BPT) exercise. A second purpose of this study was to evaluate the occurrence of side effects following caffeine consumption. It was hypothesized that 9 and 12 mg/kg of caffeine would similarly improve bar velocity during the resistance exercise, but the frequency of side effects would be dose-related.

## 2. Methods

### 2.1. Experimental Design

The study used a randomized, counterbalanced, double-blind, placebo-controlled crossover design, where each athlete acted as their own control. The randomization was performed by a member of the research team who was not involved in the data collection; thus, after assignment to interventions, both athletes and researchers were blinded to the trials. Athletes performed a familiarization session that included the 1RM bench press assessment and execution of several repetitions of maximal BPT exercise on a Smith machine with a load of 30% 1RM [27]. Three identical experimental sessions followed, with a one-week interval between sessions to allow complete recovery and ensure substance wash-out. During the three identical experimental sessions, athletes either ingested a placebo (PLAC) or 9 mg/kg of caffeine (CAF-9) or 12 mg/kg of caffeine (CAF-12). Caffeine and the placebo were administered orally 60 min before the onset of the exercise protocol [28], and at least 3 h after the last meal, to maintain the same time of absorption. After 60 min of substance intake, athletes underwent five sets with two repetitions of the BPT, while the bar was loaded with 30% of their 1RM, as measured in the pre-experimental session. Caffeine was provided in the form of capsules containing the individual dose of caffeine (Olimp Laboratories, Dębica, Poland). The manufacturer of the caffeine capsules also prepared identical placebo capsules filled out with an all-purpose flour. The study protocol was approved by the by the University Ethics Committee in accordance with the latest version of the Declaration of Helsinki. This protocol was carried out during the experimental trials performed without any deviation.

### 2.2. Study Participants

To calculate the sample size, a statistical software (G*Power, Dusseldorf, Germany) was used with the following parameters: analysis of variance with repeated measures and within factors comparison as the statistical test, an expected effect size (ES) for bar velocity equal to 0.25, an alpha level at 0.05, a statistical power at 80%, correlation among repeated measures set at r = 0.85, and three experimental conditions. The power analysis indicated that a sample of at least 10 participants was required for this study. To account for potential drop-outs, twelve healthy strength-trained male athletes (Table 1) were recruited and volunteered after completing a written consent form. The inclusion criteria were as follows: (a) free from neuromuscular and musculoskeletal disorders, (b) “resistance-trained,” defined as having a minimum of two years of resistance training experience and being able to lift at least 120% of body mass in the bench press exercise, (c) chronic caffeine intake to produce at least moderate habituation to caffeine (according to previously proposed classification) [29], and (d) previous experience in pre-workout caffeine use at a dose of 6 mg/kg with a low frequency of negative side effects. Athletes were excluded if they reported (a) a positive smoking status, (b) a potential allergy to caffeine, or (c) using any medications, dietary supplements or ergogenic aids which could potentially affect the study outcomes (e.g., beta-alanine, creatine). Habitual caffeine intake was measured by using a modified version of the validated questionnaire by Bühler et al. [30] that recorded the type and amount of caffeine-containing foods and dietary supplements. Habitual caffeine intake was recorded for the 4 weeks before the start of the experiment, following previous recommendations [29].

### 2.3. Standarizations

Athletes were instructed to maintain their usual dietary patterns and training routines during the study period and refrained from strenuous exercises 24 h before all experimental sessions. Athletes were asked to replicate their diet every 24 h before testing, recorded all the food and drinks ingested before the first experimental trial and replicated this diet in the subsequent trials. The amount of calories and the proportion of macronutrients were calculated by a qualified sports nutritionist from a 24-h diet recall (Table 1). Athletes were also asked to refrain from caffeine intake 24 h before each trial and by the end the experimental day. Adherence to these requirements was verified via a brief questionnaire administered prior to each trial before data collection.

### 2.4. Familiarization Session and One Repetition Maximum Test

During the familiarization session, athletes arrived at the laboratory at the same time of day as in the upcoming experimental sessions (between 10:00 and 11:00 a.m.). The warm-up protocol included 5 min of cycling on a stationary ergometer followed by a general upper body warm-up. Next, athletes performed 15, 10, 5, and 3 repetitions of the bench press exercise using loads corresponding to 20, 40, 60 and 80% of their estimated 1RM. The participants executed a single repetition with a constant tempo of movement (2 s duration of the eccentric phase and maximum velocity in the concentric phase, with no pause in-between) on a free barbel bench press exercise. Athletes then performed single repetitions of the bench press exercise with a 5-min rest interval between successful trials. The load for each following attempt was increased by 2.5 to 10 kg, and the process was repeated until failure. Hand placement on the barbell was individually selected (~150% individual bi-acromial distance). Ten minutes after completing the 1RM test, after instructions regarding the correct technique of BPT, the athletes executed several repetitions of maximal BPT exercise on a Smith machine with a load of 30% 1RM with a maximal tempo of movement [27].

### 2.5. Experimental Protocol

All experimental trials were conducted between 10:00 and 11:00 in the morning to avoid the effects of circadian rhythm on the outcomes of the investigation. After the warm-up procedures, which were the same as in the familiarization trial, the athletes performed five sets of two BPT repetitions at 30% 1RM on the Smith machine. The repetitions were performed with maximal tempo of movement, (the participants were encouraged to produce maximal velocity during both the eccentric and concentric phase of the BPT movement) with a 3-min rest interval between sets. Two spotters caught and lowered the loaded bar to ensure safety. A rotatory encoder (Tendo Power Analyzer, Tendo Sport Machines, Trencin, Slovakia) was used for instantaneous recording of bar velocity during the whole range of motion. During each BPT, peak bar velocity (peak velocity, in m/s); and mean bar velocity (mean velocity, in m/s) were registered. Mean velocity was obtained as the mean of the two repetitions, while peak velocity was obtained from the peak value of the best repetition.

### 2.6. Side Effects and Assessment of Blinding

Immediately after finishing testing, and after 24 h of testing, athletes were asked about their feelings associated with typical caffeine-induced side effects by using a questionnaire (nine-item measure with a yes/no response) [15,16,31]. Additionally, athletes reported if they were able to identify whether they ingested caffeine or placebo.

### 2.7. Statistical Analysis

Data are presented as the mean ± SD for performance variables and as frequency for the prevalence of side effects. All performance variables presented a normal distribution according to the Shapiro-Wilk test. Verification of differences in peak bar velocity (peak velocity), and mean bar velocity (mean velocity) was performed using a two-way (3 × 5; substance × set) analysis of variance (ANOVA) with repeated measurements. In the event of a significant main effect, post hoc comparisons were conducted using the Tukey’s test. Percent changes and 95% confidence intervals were also calculated. Effect sizes (Cohen’s *d*) were reported where appropriate and interpreted as large (*d* ≥ 0.80); moderate (*d* between 0.79 and 0.50); small (*d* between 0.49 and 0.20); and trivial (*d* < 0.20) [32]. A Fisher’s Exact test in a contingency table was conducted to evaluate whether the size of dose was associated with the occurrence of side effects. The two variables were caffeine dose with three levels (placebo, CAF-9, CAF-12) and occurrence of side effects with two levels (yes and no). Moreover, a Cochran’s Q test with pairwise comparison was conducted to evaluate differences between doses in the occurrence of side effects. The magnitude of association between caffeine dose and the occurrence of side effects was described by Cramer’s V with the following approach: low (between 0.1 and 0.3), moderate (between 0.3 and 0.5) and high (>0.5). The significance level was set at *p* < 0.05 for all statistical analysis.

## 3. Results

### 3.1. Performance

The two-way repeated measures ANOVA indicated a significant main effect of substance for peak velocity (F = 9.12; *p* < 0.01) and for mean velocity (F = 8.79; *p* < 0.01). Post hoc analyses for main effect of substance indicated significant increases in peak velocity after the intake of CAF-9 (*p* < 0.01; ES = 0.36) and CAF-12 (*p* < 0.01; ES = 0.33) compared to PLAC (Table 2). The intake of CAF-9 (*p* < 0.01; ES = 0.42) and CAF-12 (*p* < 0.01; ES = 0.42) also increased mean velocity compared to PLAC. There were no significant differences in peak velocity (*p* = 0.91) and mean velocity (*p* = 0.96) between the two doses of caffeine. The two-way repeated measures ANOVA indicated no significant substance × set main interaction effect for peak velocity (F = 0.56; *p* = 0.81) and mean velocity (F = 0.72; *p* = 0.67). The results of mean velocity and peak velocity in individual sets for PLAC, CAF-9 and CAF-12 conditions are presented in Table 3.

### 3.2. Side Effects and Assessment of Blinding

In the assessment of the prevalence of side effects immediately after the end of testing, the Fisher’s Exact Test showed a statistically significant association between the caffeine dose and anxiety or nervousness (*p* = 0.001; Cramer’s V = 0.617) and moderate association of dose with headache (*p* = 0.032; Cramer’s V = 0.455), increased vigor/activeness (*p* = 0.028; Cramer’s V = 0.478), and perception of performance improvement (*p* = 0.075; Cramer’s V = 0.403). Twenty-four hours after testing, statistically significant associations between caffeine dose and tachycardia and heart palpitations (*p* < 0.001; Cramer’s V = 0.764), anxiety or nervousness (*p* = 0.001; Cramer’s V = 0.575), headache (*p* < 0.001; Cramer’s V = 0.727), increased vigor/activeness (*p* = 0.006; Cramer’s V = 0.533, and insomnia (*p* = 0.009; Cramer’s V = 0.519), were observed. Table 4 shows details of the differences determined by Cochran’s Q test and percentage frequency of the side effects in all three experimental trials, as assessed immediately after, and for 24 h after, the test protocol.

Only 33% of the sample reported that they ingested caffeine when they were given the placebo, while the remaining 92% and 100% correctly guessed caffeine trials after intake CAF-9 and CAF-12, respectively.

## 4. Discussion

The main finding of the study was that the acute intake of high doses of caffeine (i.e., 9 and 12 mg/kg) enhanced mean and peak bar velocity during a testing protocol that included five sets of two repetitions of the BPT exercise in athletes habituated to chronic caffeine intake. Specifically, the observed ergogenic benefits were present for mean and peak velocity and for both doses of caffeine when compared to a placebo situation, while the performance benefits were of similar magnitude between caffeine doses. However, the frequency of side effects occurring immediately after the experimental trials, and during the following 24 h, was significantly higher after the intake of CAF-12, while CAF-9 produced minimal side-effects. Therefore, the outcomes of this investigation indicate that high doses of caffeine (from 9 to 12 mg/kg) are effective in increasing mean and peak velocity during the BPT in athletes habituated to caffeine. However, the selection of 9 mg/kg is recommended for athletes habituated to caffeine because this dosage produced an ergogenic effect on this type of resistance exercise as high as the 12 mg/kg dose but with minimal prevalence of side effects.

Previous research has shown that acute caffeine intake, of doses ranging from 3 to 9 mg/kg, increases strength-power performance during different forms of resistance exercise [16,27,33,34,35,36], which is in line with the results of this study. Additionally, a previous study confirmed the beneficial impact of 11 mg/kg of caffeine on maximum strength performance [15]. However, to the best of our knowledge, this is the first investigation that considered the acute impact of 12 mg/kg of caffeine during resistance exercise. The study of high doses of caffeine may be important for athletes habituated to caffeine because evidence points toward the use of high amounts of caffeine in some athletes to overcome tolerance to the ergogenic effect of caffeine developed by chronic intake. For this reason, investigations with high doses of caffeine on resistance exercise have used athletes with prior habituation to caffeine. The results of the presented study showed that both doses of caffeine (9 and 12 mg/kg) significantly increased bar velocity during the BPT compared to the PLAC condition in athletes habituated to caffeine. Interestingly, the overall increase in bar velocity, when compared to PLAC, was of similar magnitude with the intake of 9 and 12 mg/kg of caffeine with no differences between the doses. Therefore, increasing the acute intake of caffeine to 12 mg/kg did not result in an additional enhancement of physical performance compared to 9 mg/kg. A previous study by Wilk et al. [15] also showed a significant improvement of strength performance (1RM bench press test) after the intake of 9 and 11 mg/kg of caffeine, with no differences between the doses. Collectively, these data suggest that there are no additional acute benefits of consuming caffeine above a dose of 9 mg/kg in athletes habituated to caffeine, which suggests the existence of possible limits for maximum caffeine intake in habituated participants.

A previous study used an experimental protocol that included the same protocol of exercise (BPT exercise; 30% 1 RM; five sets of two reps), resistance trained athletes habituated to caffeine and two doses of caffeine (3 and 6 mg/kg) [27]. In that investigation, the ingestion of either 3 and 6 mg/kg of caffeine was effective in enhancing mean bar velocity, while the magnitude of the ergogenic effect was very similar to the one reported here with 9 and 12 mg/kg of caffeine [27]. However, in the prior investigation, 3 and 6 mg/kg of caffeine were ineffective in enhancing peak bar velocity, contrary to the benefits obtained with 9 and 12 mg/kg. Therefore, it may be assumed that for resistance training exercise with the intention of enhancing maximal power output, it may be necessary to consume at least 9 mg/kg of caffeine in athletes habituated to caffeine.

Previous research has suggested that overall increase of performance after acute caffeine intake is associated with the participant’s level of habituation to caffeine [22]. It should be noted that in the presented research the overall increase in peak and mean velocity occurred after the intake of 9 and 12 mg/kg of caffeine for moderate-to-high caffeine users (5.3 ± 1.4 mg/kg/day). Although several previous studies found positive acute effects of 3-to-6 mg/kg of caffeine in habituated participants [37,38], the investigations were carried out on low to mild caffeine consumers. For example, Grgic and Mikulic [38] found an increase in mean and peak velocity during the bench press exercise after the intake of 3 mg/kg of caffeine in a group consuming 235 ± 82 mg/day (~2.8 mg/kg/day) of caffeine per day. Similarly, Sabol et al. [37] found an improvement in medicine ball throw distance in a study conducted on habituated to caffeine participants (358 ± 210 mg/day; ~4.1 mg/kg/day) after ingestion of 6 mg/kg of caffeine. However, in the study of Sabol et al. [37], lower caffeine doses (2 and 4 mg/kg of caffeine) did not increase performance, which may suggest a reduction in ergogenic effects of caffeine in subjects with higher daily caffeine consumption. Interestingly, previous studies [14,27,36] conducted on participants with daily caffeine intakes similar to those in the presented research (range from 4.2 to 5.1 mg/kg/day) did not show a positive effect of 3 and 6 mg/kg of caffeine (doses similar to a daily level of caffeine consumption) [14], or only partial improvement of strength-power performance were observed [27,36]. A summary of all these data suggests that in athletes with low habituation to caffeine (2-to-4 mg/kg/day), the ingestion of an acute dose of 3 and 6 mg/kg of caffeine may exert ergogenic benefits in resistance and power-based exercise. However, in athletes with moderate-to-high habituation to caffeine (4-to-6 mg/kg/day), doses of 9 mg/kg may be needed to obtain such ergogenic benefits on resistance-based exercise.

It should be noted that consumption of high doses of caffeine (≥9 mg/kg), although ergogenic for athletes habituated to chronic intake of caffeine, present higher prevalence of caffeine-associated side effects [22]. In the present study, the intake of 12 mg/kg of caffeine increased the prevalence of tachycardia/palpitations events, anxiety and activeness, headache, and gastrointestinal discomforts just after the end of the exercise protocol, while most of these drawbacks were still persistent 24 h after testing (Table 4). Interestingly, the current research showed that the prevalence of these side effect was dose-dependent, as 9 mg/kg produced slightly higher, and in most cases non-statistically significant prevalence of side effects when compared to the placebo. The relatively low frequency of side effects observed in the trial with 9 mg/kg of caffeine can be attributed to the fact that habitual caffeine users may develop tolerance to certain physiological effects of caffeine [21]. It should be noted that participants involved in the presented study had significantly higher daily caffeine intake than those from previous studies [16,17,18,19], which may have promoted the low prevalence of side effects with 9 mg/kg. Thus, the reported level of side effects can be attributed to the fact that with repeated and regular intake of caffeine, the dose needed to induce caffeine’s physiological effects increases [21]. The results of the presented study suggest that for moderate-to-high caffeine users, 9 mg/kg may be relatively safe, taking into account that dose produced a high prevalence of side effects in low consumers of caffeine [16]. However, despite expected performance improvements, the recommendation of such doses of caffeine has to be individualized as there is the possible occurrence of side effects that may negatively impact resistance exercise performance (e.g., in case of gastrointestinal problems) and recovery (e.g., in case of insomnia) [22].

In addition to its strengths, the present study has several limitations which should be addressed. (1) The study did not include any biochemical analysis, such as plasma/urinary caffeine concentrations which could help explain the direct causes of performance changes. (2) There was no analysis of genetic intolerance to caffeine in the tested participants. (3) We assed only several side effects 24 h after caffeine ingestion and the exercise protocol and thus long-term effects are unknown. This is important as evidence suggests that the prevalence of side effects increases with chronic intake, even with lower doses [39]. (4) During trials with caffeine, most participants reported that they had ingested caffeine. However, this might be associated with using higher doses of caffeine than used in previous studies, where a similar blinding protocol masked caffeine use in trials with lower doses [37,38]. (5) We analyzed only the effects of caffeine intake in moderate to high caffeine consumers; therefore, generalizing these results to a population with other levels of caffeine consumption would not be correct. The extrapolation of these conclusions to individuals with lower habitual caffeine intake, and for long-term use, should not be done because it may produce higher prevalence of side effects that may affect athletes’ performance and well-being.

## 5. Conclusions

The intake of either 9 and 12 mg/kg of caffeine 60 min before a session with several sets of BPT exercise increased mean and peak bar velocity in athletes with moderate-to-high habitual caffeine intake. Both doses of caffeine produced a similar performance benefit but the ingestion of 12 mg/kg of caffeine significantly increased the frequency of negative side effects immediately after, and for the 24 h after testing. Thus, the ingestion of 9 mg/kg of caffeine prior to ballistic exercise can be considered an effective and safe supplementation protocol for individuals habituated to caffeine.

## Figures and Tables

**Table 1 jcm-10-04380-t001:** Participants’ characteristics.

Variable [units]	Mean ± Standandard Deviation (*n* = 12)
Age (years)	25.2 ± 1.3
Body mass (kg)	85.4 ± 13.2
Height (cm)	180.6 ± 4.4
Body Fat (%)	12.1 ± 3.0
Resistance training experience (years)	4.1 ± 1.3
Bench press exercise 1RM (kg)	121.1 ± 30.5
Ratio of 1 bench press exercise to body mass (%)	140.6 ± 15.0
Habitual caffeine intake (mg/kg/day)	5.3 ± 1.4
Habitual caffeine intake (mg/day)	463.3 ± 171.3
Energy intake (kcal/day)	3341.8 ± 568.8
Protein (% of total energy intake)	19.5 ± 3.9
Fat (% of total energy intake)	28.3 ± 2.3
Carbohydrates (% of total energy intake)	52.3 ± 4.0

1RM—one-repetition maximum.

**Table 2 jcm-10-04380-t002:** Average values of peak and mean bar velocity during five sets of the bench press throw with the ingestion of 9 and 12 mg/kg of caffeine or with a placebo in resistance trained athletes habituated to caffeine.

Bench Press Throw	Conditions			*p* for Main Effect of Substance
	PLAC	CAF-9	CAF-12	
Peak bar velocity [m/s]	2.17 ± 0.19	2.24 ± 0.20	2.23 ± 0.17	<0.01
Mean bar velocity [m/s]	1.37 ± 0.10	1.41 ± 0.09	1.41 ± 0.09	<0.01

Data represents mean values of the five sets. All data are presented as mean ± standard deviation. PLAC: placebo; CAF-9: caffeine 9 mg/kg; CAF-12: caffeine 12 mg/kg.

**Table 3 jcm-10-04380-t003:** Peak and mean var velocity for each of the five sets of the bench press throw with the ingestion of 9 and 12 mg/kg of caffeine or with a placebo in resistance trained athletes habituated to caffeine.

Conditions		Set 1	Set 2	Set 3	Set 4	Set 5
**Peak Bar Velocity [m/s]**						
PLAC (95%CI)		2.14 ± 0.16 (2.04 to 2.25)	2.17 ± 0.19 (2.04 to 2.29)	2.20 ± 0.19 (2.08 to 2.33)	2.18 ± 0.20 (2.05 to 2.31)	2.16 ± 0.22 (2.02 to 2.31)
CAF-9 (95%CI)		2.21 ± 0.21 (2.07 to 2.35)	2.22 ± 0.19 (2.09 to 2.34)	2.26 ± 0.19 (2.13 to 2.38)	2.24 ± 0.18 (2.13 to 2.36)	2.28 ± 0.22 (2.13 to 2.43)
CAF-12 (95%CI)		2.20 ± 0.14 (2.10 to 2.29)	2.23 ± 0.17 (2.11 to 2.34)	2.24 ± 0.17 (2.13 to 2.35)	2.24 ± 0.18 (2.12 to 2.36)	2.26 ± 0.19 (2.13 to 2.39)
ES	PLAC vs. CAF-9	0.37 (−0.44, 1.17)	0.26 (−0.55, 1.06)	0.32 (−0.50, 1.11)	0.32 (−0.50, 1.11)	0.55 (−0.29, 1.34)
	PLAC vs. CAF-12	0.40 (−0.42, 1.19)	0.33 (−0.48, 1.13)	0.22 (−0.59, 1.02)	0.32 (−0.59,1.01)	0.49 (−0.90, 0.70)
	CAF-9 vs. CAF-12	0.06 (−0.85, 0.75)	0.06 (−0.75, 0.85)	0.01 (−0.91, 0.69)	0.00 (−0.80, 0.80)	0.01 (−0.89, 0.71)
**Mean bar velocity [m/s]**						
PLAC (95%CI)		1.34 ± 0.07 (1.29 to 1.39)	1.37 ± 0.12 (1.30 to 1.45)	1.38 ± 0.09 (1.32 to 1.44)	1.38 ± 0.10 (1.31 to 1.45)	1.36 ± 0.11 (1.28 to 1.44)
CAF-9 (95%CI)		1.42 ± 0.12 (1.34 to 1.50)	1.40 ± 0.07 (1.36 to 1.45)	1.40 ± 0.10 (1.34 to 1.47)	1.43 ± 0.08 (1.38 to 1.48)	1.42 ± 0.08 (1.36 to 1.47)
CAF-12 (95%CI)		1.39 ± 0.09 (1.33 to 1.44)	1.41 ± 0.06 (1.37 to 1.45)	1.42 ± 0.08 (1.36 to 1.47)	1.42 ± 0.10 (1.35 to 1.49)	1.41 ± 0.09 (1.35 to 1.47)
ES	PLAC vs. CAF-9	0.81 (−0.04,1.62)	0.31 (−0.51, 1.10)	0.21 (−0.60, 1.01)	0.55 (−0.28, 1.35)	0.62 (−0.22, 1.42)
	PLAC vs. CAF-12	0.62 (−0.22, 1.42)	0.42 (−0.40, 1.22)	0.47 (−0.36, 1.26)	0.40 (−0.42, 1.19)	0.50 (−0.33, 1.29)
	CAF-9 vs. CAF-12	0.28 (−1.08, 0.53)	0.15 (−0.65, 0.95)	0.22 (−0.59, 1.02)	0.11 (−0.91, 0.69)	0.12 (−0.91, 0.69)

All data are presented as mean ± standard deviation. CI: confidence interval. PLAC: placebo; CAF-9: caffeine 9 mg/kg; CAF-12: caffeine 12 mg/kg. ES: effect size.

**Table 4 jcm-10-04380-t004:** Frequency of side effects immediately after, and 24 h after, a bench press throw session with the ingestion of 9 and 12 mg/kg of caffeine or with a placebo in resistance trained athletes habituated to caffeine.

	PLAC		CAF-9		CAF-12	
	Just after	24 h after	Just after	24 h after	Just after	24 h after
Increased urine output	8%	0%	25%	33% *	25%	25%
Tachycardia and heart palpitations	17%	0%	33%	17%	67% †	83% †#
Anxiety or nervousness	6%	0%	42%	33%	83% †	67% †
Headache	17%	0%	25%	25%	67% †#	83% †#
Gastrointestinal problems	0%	0%	17%	25%	33% †	25%
Increased sweating	8%	0%	33%	25%	33%	25%
Increased vigor/activeness	25%	0%	75%*	25%	75% †	58% †
Perception of performance improvement	25%	-	67%*	-	25% #	-
Insomnia	-	0%	-	33%	-	58% †

Data are presented as the frequency of affirmative responses to the existence of side effects. * Significant difference (*p* < 0.05) between CAF-9 and PLAC. † Significant difference (*p* < 0.05) between CAF-12 and PLAC. # Significant difference (*p* < 0.05) between CAF-9 and CAF-12.

## Data Availability

The datasets used and/or analyzed during the current study are available from the corresponding author on reasonable request.

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
