# Peer review of "Acute Effects of High Doses of Caffeine on Bar Velocity during the Bench Press Throw in Athletes Habituated to Caffeine: A Randomized, Double-Blind and Crossover Study"

_jcm, 2021, doi:10.3390/jcm10194380_

Round 1
Reviewer 1 Report
The article is of interest, and it is well designed, executed and written.
Some details are missing to comply with the CONSORT guidelines:
- Title: study design should be indicated
- Protocol registration: It would be necessary (editors should reflect on whether they will start to ask for it as an obligation or not) that the authors had registered a protocol before the start of the study, since it is the only way to verify if there have been changes to the study with respect to the protocol. The reason to ask abou that is to verify point 3b of the CONSORT guidelines: "Important changes to methods after trial commencement (such as eligibility criteria), with reasons".
- Methods: Elibiility criteria, research procedures, oucomes measures, sample size are clear.
- More information is needed about the method used to generate the random allocation sequence, type of randomisation, mechanism used to implement the random allocation sequence, and who assigned participants to interventions.
- Methods: Also more information is needed about who was blinded after assignment to interventions .
- Results - Tables: include number of participants analised
- Results: are well described.
- Discussion: well written.
Author Response
Response to Reviewers’ comments
Acute Effects of High Doses of Caffeine on Bar Velocity during the Bench Press Throw in Athletes Habituated to Caffeine: A Randomized, Double-Blind and Crossover Study
We sincerely thank the Reviewers and the Editors of Journal of Clinical Medicine for careful peer-reviewing the manuscript and for their valuable comments provided in the review letter. Below, we have included a response letter where we have replied item-by-item to the comments provided by the Reviewers. We have highlighted the changes within the manuscript in yellow. We feel that the manuscript is improved in the light of the suggested changes.
Reviewer #1:
The article is of interest, and it is well designed, executed and written.
Reply - Thank you for the evaluation and for the comments provided in this review letter.
Some details are missing to comply with the CONSORT guidelines:
Title: study design should be indicated
Reply - Thank you for this comment. According to your suggestion we have indicated the study design in the title.
Protocol registration: It would be necessary (editors should reflect on whether they will start to ask for it as an obligation or not) that the authors had registered a protocol before the start of the study, since it is the only way to verify if there have been changes to the study with respect to the protocol. The reason to ask about that is to verify point 3b of the CONSORT guidelines: "Important changes to methods after trial commencement (such as eligibility criteria), with reasons".
Reply - Thank you for this comment. We have followed CONSORT guidelines to prepare the manuscript and that there was not any change in the methods after trial commencement. We have indicated this in the manuscript.
Methods: Elibiility criteria, research procedures, oucomes measures, sample size are clear.
More information is needed about the method used to generate the random allocation sequence, type of randomisation, mechanism used to implement the random allocation sequence, and who assigned participants to interventions.
Reply – Thank you for this clever comment. The randomization was performed by a member of the research team who was not involved in the data collection (M.W.) .The order of trials (PLAC, CAF-9, CAF-12) was chosen randomly and in a counterbalance order using free online randomization program (randomization.com).
Methods: Also more information is needed about who was blinded after assignment to interventions.
Reply – Thank you for this comment. According to your suggestions we added this information as following ‘thus after assignment to interventions both, athletes and researchers were blinded to the trials’.
Results - Tables: include number of participants analised.
Reply – Thank you for this comment. According to your suggestions we have included number of participants.
Results: are well described.
Discussion: well written.
Reply – Than you for this positive comment.
Reviewer 2 Report
Thank you very much for your manuscript submission. It is well-written, well-supported, and you report some interesting findings with practical application considerations for resistance-based athletes. Some specific comments:
Abstract:
- Remove headings (i.e. Purpose, Methods)
Introduction:
- Page 2: “It should be noted that such high doses of caffeine in these three studies [14–16,21] leaded to…” – change to “led to”
- Page 2: “Athletes performed a familiarization session that included the assessment 1RM in the bench press exercise” – change to “that included the 1RM bench press assessment. Three identical experimental sessions followed with a one-week interval between sessions…and ensure substances wash-out.”
Methods:
- Pages 2/3: include references for 1) caffeine being administered 60 minutes prior to exercise and 2) 30% bar load
- Did the familiarization session include the bar throw or just the 1RM? I see later in the methods it did, so I recommended that general info be added in the experimental section
- Table 1: remove the bold style for age; create an additional row for caffeine intake (mg/day). It would be beneficial visually to have them on two separate rows, rather than next to each other in the same row
- Experimental protocol: change to “10:00 and 11:00”
- Please add references to support your procedures in the familiarization and experimental protocol sections
- How was the placebo created? Details on this should be provided.
- Page 5: Delete “;” after trivial
Results:
- Table 2 is not referenced in the text. Begin the results section with this finding, then move into the substance x set interaction results (table 3)
- Table 3: bold “mean bar velocity [m/s]. PLAC set 3 is missing a “)” . Can you add ES under 95% CI? The table looks a bit mis-formatted with the ES at the bottom and everything shifted over.
- Page 6: change to “Only 33% of the sample reported they they ingested…”

Author Response
Response to Reviewers’ comments
Acute Effects of High Doses of Caffeine on Bar Velocity during the Bench Press Throw in Athletes Habituated to Caffeine: A Randomized, Double-Blind and Crossover Study
We sincerely thank the Reviewers and the Editors of Journal of Clinical Medicine for careful peer-reviewing the manuscript and for their valuable comments provided in the review letter. Below, we have included a response letter where we have replied item-by-item to the comments provided by the Reviewers. We have highlighted the changes within the manuscript in yellow. We feel that the manuscript is improved in the light of the suggested changes.
Reviewer #2:
Thank you very much for your manuscript submission. It is well-written, well-supported, and you report some interesting findings with practical application considerations for resistance-based athletes. Some specific comments:
Reply - Thank you for this evaluation and for the comments provided in this review letter.
Abstract:
Remove headings (i.e. Purpose, Methods)
Reply – Thank you for this comment. Headings has been removed.
Introduction:
Page 2: “It should be noted that such high doses of caffeine in these three studies [14–16,21] leaded to…” – change to “led to”
Reply – Thank you for this comment. According to your suggestions we have corrected it.
Page 2: “Athletes performed a familiarization session that included the assessment 1RM in the bench press exercise” – change to “that included the 1RM bench press assessment. Three identical experimental sessions followed with a one-week interval between sessions…and ensure substances wash-out.”
Reply – Thank you for this comment. According to your suggestions we have rewritten this sentence.
Methods:
Pages 2/3: include references for 1) caffeine being administered 60 minutes prior to exercise and 2) 30% bar load
Reply – Thank you for this comment. According to your suggestions we have added references to support this procedure.
Did the familiarization session include the bar throw or just the 1RM? I see later in the methods it did, so I recommended that general info be added in the experimental section
Reply – Thank you for this comment. According to your suggestions we have added this information in the experimental section.
Table 1: remove the bold style for age; create an additional row for caffeine intake (mg/day). It would be beneficial visually to have them on two separate rows, rather than next to each other in the same row
Reply – Thank you for this comment. According to your suggestions we have corrected the table.
Experimental protocol: change to “10:00 and 11:00”
Reply – Thank you for this comment. According to your suggestions we have corrected it.
Please add references to support your procedures in the familiarization and experimental protocol sections
Reply – Thank you for this comment. According to your suggestions we have added references.
How was the placebo created? Details on this should be provided.
Reply – Thank you for this comment. The manufacturer of the caffeine pills also prepared identical placebo capsules filled out with an all-purpose flour.
Page 5: Delete “;” after trivial
Reply – Thank you for this comment. According to your suggestions we have deleted it.
Results:
Table 2 is not referenced in the text. Begin the results section with this finding, then move into the substance x set interaction results (table 3).
Reply – Thank you for this comment. According to your suggestions we have corrected results section.
Table 3: bold “mean bar velocity [m/s]. PLAC set 3 is missing a “)” . Can you add ES under 95% CI? The table looks a bit mis-formatted with the ES at the bottom and everything shifted over.
Reply – Thank you for this comment. According to your suggestions we have formatted Table and added ES under 95% CI.
Page 6: change to “Only 33% of the sample reported they they ingested…”
Reply – Thank you for this comment. According to your suggestions, we have corrected it.